Study on the inhibitory mechanism of fig leaf extract against postharvest Fusarium in melon

Yao Jun 1 2
Geng Xinli 2 89069899@qq.com
Zheng Heyun 2
Wang Zhiwei 2
Zhang Cuihuan 2
Li Jing 2
Maimaiti Zaituna 2
Qin Yong 1 xjndqinyong@sina.com
1 College of Horticulture, Xinjiang Agricultural University , Xinjiang, Urumqi , China
2 Xinjiang Uygur Autonomous Region Grapevine and Melon Fruit Research Institute , Xinjiang, Shanshan , China
Yasin Nasim
Electronic publication date: 2024 Jul 25
Publication date: 2024
Volume: 12
Electronic Location ID: e17654
Received 2024 Jan 30; Accepted 2024 Jun 7
Copyright: © 2024 Yao et al.
Copyright year: 2024
Copyright holder: Yao et al.
License: This is an open access article distributed under the terms of the Creative Commons Attribution License, which permits unrestricted use, distribution, reproduction and adaptation in any medium and for any purpose provided that it is properly attributed. For attribution, the original author(s), title, publication source (PeerJ) and either DOI or URL of the article must be cited.
License URL: https://creativecommons.org/licenses/by/4.0/

Keywords: Fig leaf extract, Fusarium, Mechanism of inhibition, Broadly targeted metabolomics, Melon

Funding: Nature Found projects of Institute in Xinjiang 2022D01A118 Agricultural Industrial Technology System of China CARS-25 XinJiang Agriculture Research System XJARS-06 “Three Rural” Backbone Talent Training Programme 2022SNGGGCO010 This work was supported by the Nature Found projects of Institute in Xinjiang (2022D01A118), Agricultural industrial technology system of China (CARS-25), and the XinJiang Agriculture Research System (XJARS-06). Xinjiang “Three Rural” Backbone Talent Training Programme: 2022SNGGGCO010. The funders had no role in study design, data collection and analysis, decision to publish, or preparation of the manuscript.

==============================
The objective of this study was to explore the fungistatic mechanism of fig leaf extract against Fusarium and to provide a theoretical basis for the development of new plant-derived fungicides.

Methods

The fungistaticity of fig leaf extract were analyzed by the ring of inhibition method. Fusarium equiseti was selected as the target for analyzing its fungistatic mechanism in terms of mycelial morphology, ultrastructure, cell membrane permeability, membrane plasma peroxidation, reactive oxygen species (ROS) content and changes in the activity of protective enzymes. The effect of this extract was verified in melon, and its components were determined by metabolite analysis using ultraperformance liquid chromatography‒mass spectrometry (UPLC‒MS).

Results

Fig leaf extract had an obvious inhibitory effect on Fusarium, and the difference was significant (P < 0.05) or highly significant (P < 0.01). Scanning and transmission electron microscopy revealed that F. equiseti hyphae exhibited obvious folding, twisting and puckering phenomena, resulting in an increase in the cytoplasmic leakage of spores, interstitial plasma, and the concentration of the nucleus, which seriously damaged the integrity of the fungal cell membrane. This phenomenon was confirmed by propidium iodide (PI) and fluorescein diacetate (FAD) staining, cell membrane permeability and malondialdehyde (MDA) content. Fig leaf extract also induced the mycelium to produce excessive H2O2,which led to lipid peroxidation of the cell membrane, promoted the accumulation of MDA, accelerated protein hydrolysis, induced an increase in antioxidant enzyme activity, and disrupted the balance of ROS metabolism; these findings showed that fungal growth was inhibited, which was verified in melons. A total of 1,540 secondary metabolites were detected by broad-targeted metabolomics, among which the fungistatic active substances flavonoids (15.45%), phenolic acids (15%), and alkaloids (10.71%) accounted for a high percentage and the highest relative content of these substances 1,3,7,8-tetrahydroxy-2- prenylxanthone, 8-hydroxyquinoline and Azelaic acid were analysed for their antimicrobial, anti-inflammatory, antioxidant, preventive effects against plant diseases and acquisition of resistance by plants. This confirms the reason for the fungicidal properties of fig leaf extracts.

Conclusion

Fig leaf extract has the potential to be developed into a plant-derived fungicide as a new means of postharvest pathogen prevention and control in melon.

Introduction

China is one of the largest producers of Cucumis melo in the world, with 387,100 hectares accounting for more than 45% of the global total planted area and 14,071,600 tonnes of production, or approximately 50%, according to the World System of Agricultural Organisations (FAO) Database (2022). Melon has attracted much attention as a specialty cash crop. However, melon subjected to long-distance transport and short-term storage will experience a 5–15% loss in the sales process and a 3–5% loss through wholesale, distribution and other intermediate links. Studies have shown that postharvest pathogenic fungal colonies of melons are diverse, with Fusarium being the dominant fungal genus (Zheng et al., 2020). More than 15% of Brazilian melons are reportedly lost to Fusarium fruit rot during export transport (Nogueira et al., 2023). Fusarium is an important postharvest pathogen of fruits and vegetables and can infect fruits and vegetables such as blackthorn plum (Hye, Thi & Hyang, 2016), watermelon (Balasubramaniam et al., 2023), winter pumpkin (Kitabayashi et al., 2023), banana (Kanhe, Abhilasha & Girish, 2019), pineapple (Barral et al., 2019), kiwifruit (Wang et al., 2023) and melon (Nogueira et al., 2023). In addition to causing postharvest decay, Fusarium metabolically produces various mycotoxins, such as deoxynivalenol (DON) (Ferreira, Hirooka & Ferreira, 2018), zearalenone (ZEN) (Marin et al., 2004), fumonisin FB1 (FB1) (Juglal, Govinden & Odhav, 2002) and other mycotoxins, which are harmful to the safety of fruits and vegetables.

Chemical fungicides can effectively control a variety of postharvest diseases caused by Fusarium, but there are many problems, such as pollution of the environment, pesticide residues harmful to health and drug resistance. Therefore, the research and development of new natural plant-derived fungicides agents has become an important direction for food preservation. Several studies have reported that extracts of Cunoniaceae species (Fogliani, Bouraïma-Madjebi & Medevielle, 2002), Morus alba and Aegle marmelos (Naseem et al., 2021), Mesquite (López-Anchondo et al., 2021), bagasse (Hadimani et al., 2023), Clove and Thyme (Monteiro, Ferreira & Silva, 2013), and cumin seeds (Wang et al., 2017) have significant inhibitory effects on Fusarium spores. The use and residue of pesticides can be effectively reduced through the application of plant-derived extract preparations. Fig leaves are rich in a variety of functional components, such as flavonoids, polysaccharides, polyphenols and other compounds, which have antioxidant, anti-inflammatory, anticancer and antimicrobial effects (Mustafa et al., 2019; Mahmoudi et al., 2016). Figs have low leaf utilization (Teruela et al., 2021), and fig leaf extract has high phenolic and antioxidant activities (Stănică et al., 2021). Using fig leaf extract, Ustun, Onbas & Celik (2022) and Arumugam et al. (2021) synthesized nanoparticles with antioxidant, fungisatatic and photocatalytic properties. The remarkable fungistaticity of fig leaf extracts have attracted the attention of researchers. Currently, postharvest rot disease in melon caused by Fusarium is controlled by UV-C radiation (Daniel, Katia & Rosa, 2021), heat treatment (Bokshi, Morris & McConchie, 2007), bioinducer induction (Sanchez et al., 2009), and pulsed light (Francisco et al., 2020); however, plant-derived fungisatatic materials for the biological control of Fusarium in postharvest melon have rarely been reported. In this study, we used the fungisatatic materials of fig leaf extract as a plant source material to control postharvest rot disease in melon caused by Fusarium, clarify the fungisatatic mechanism of Fusarium, propose a new method for the green prevention and control of postharvest rot disease in fruits and vegetables and provide a theoretical basis for the development of a new type of plant source fungicide.

Materials and Methods

Materials

Materials for testing

Xinjiang Early Yellow fig leaves were harvested in November 2022 in Lukqin Town, Shanshan County, Xinjiang, after fruit ripening and harvesting.

Strains H-3, H-5, H-6, H-10, 1 and 2 were isolated for melon storage fungal disease testing and sequenced to obtain the ITS sequence of the strains. After a BLAST search in GenBank to obtain the registration number of similar strains, the sequence of these isolates and the similarity of the reference sequence were 99–100% (as shown in Table 1). Fusarium equiseti, a highly percentage of isolates, was selected as a target for the study of fungi inhibition mechanism.

Table 1 Results of identification of fungal pathogens.

Number of isolates	Accession numbers of reference strain	Identified species	Sequence similarity (%)	
H-3	MT974000.1	Fusarium equiseti	99.80	
H-5	AB369435.1	Fusarium chlamydosporum	100.00	
H-6	KU528847.1	Fusarium oxysporum	99.81	
H-10	OQ818125.1	Fusarium solani	99.25	
1	OQ555140.1	Fusarium equiseti	99.61	
2	MK883240.1	Fusarium solani	99.63	

Test instruments

An ultrasonic cell pulverizer (Ningbo Xinzhi Biotechnology Co., Ningbo, China), a rotary evaporator (Shanghai Yarong Biochemical Instrument Factory., Shanghai, China), an ultraclean bench (Biobase Holdings Group Ltd., Shandong, China), a inverted fluorescence microscope (Olympus Corporation., Tokyo, Japan), an autoclave (Japan Sanyo Co., Sapporo-shi, Japan), an ultraviolet spectrophotometer (Eppendorf AG, Hamburg, Germany), a high-speed low-temperature centrifuge (Eppendorf AG, Hamburg, Germany), a constant-temperature shaker (Shanghai Lingyi Industry Co., Shanghai, China) and UPLC-ESI-MS/MS system (UPLC, ExionLC™ AD, https://sciex.com.cn/; MS, Applied Biosystems 6500 Q TRAP, https://sciex.com.cn/). were used.

Experimental methods

Preparation of fig leaf extracts

The preparation method was performed according to the methods of Zhao et al. (2018), with slight adjustment. The fig leaves were washed after harvesting, dried naturally, placed in a 60 °C drying oven, crushed, and sieved through 80 mesh to obtain a dry powder. Fig leaf powder (5.0 g) was weighed into a 150 mL conical flask, 100 mL 70% ethanol solution was added, and the powder was soaked at room temperature for 12 h. The mixture was put into the ultrasonic cell pulverizer for 30 min. The ethanol extracts of fig leaves were filtered with filter paper, the filtrate was put in a rotary evaporator at 70 °C and distilled under reduced pressure to obtain the fig leaf concentrate master batch, which was sealed, and placed in a 4 °C refrigerator for preservation.

Antifungal potential of fig leaf extracts

In the growth rate assay (Chen, 1990), the master leaf extract was mixed with potato dextrose agar (PDA) medium at a ratio of 5:1 to create a drug-carrying medium, sterile water was used as a blank control, and each treatment was repeated five times. The isolated and purified H-3, H-5, H-6, H-10, 1 and 2 strains were activated in PDA media at 28 °C for 72 h, after which the fungi were removed from the edge of the colony with a perforator to make a pellet with a diameter of 6 mm. The resulting pellet was subsequently transferred to the drug-loaded PDA Petri dishes and blank control dish and incubated at 28 °C for 72 and 96 h. The diameter of the colonies was subsequently measured via the crossover method, after which the inhibition rate of the growth of mycelia was calculated. The relevant formulae were as follows:

Mycelial growth inhibition rate (%) = (mycelial growth diameter of control group − mycelial growth diameter of treatment group)/(mycelial growth diameter of control group − diameter of mycelial cake) × 100.

Effect of fig leaf extract on F. equiseti cell morphology

In the ultraclean bench with 6 mm sterilized punch in the edge of the activated F. equiseti fungi, with a sterilized inoculation needle to pick up complete colonies, inverted to 100 mL potato liquid culture medium, placed in a constant temperature shaker at 28 °C incubated for 24 h, and then combined with 20 mL of sterile water as a control or 20 mL of concentrated fig leaf extract as a treatment. The potato glucose liquid medium containing the fig leaf extract was incubated for 24 h, the organisms were collected and washed with phosphate-buffered saline (pH 7.2) three times, and the treated samples were put into 3% glutaraldehyde fixative and placed at 4 °C for 12 h fixation. The samples were subsequently sent to Guangzhou Jingyan Testing Technology Service Co. for electron microscopic observation.

Effect of fig leaf extract on the cell membrane permeability of F. equiseti

PI and FAD staining

Propidium iodide (PI) was added to a solution at a concentration of 400 µg/mL in PBS, and fluorescein diacetate (FAD) was dissolved in acetone and added to a solution at a concentration of 5 mg/mL. When staining, the spores of F. equiseti were made into a 1 × 106 spore suspension, which was added to the PI or FAD solution and then left in the dark at room temperature for 5 min for staining. Staining was observed and photographed under a inverted fluorescence microscope.

Determination of extracellular conductivity

Mycelia of F. equiseti samples cultured in a constant temperature shaker at 28 °C for 24 h, were, under aseptic conditions, added to a 50 mL sterile centrifuge tube, centrifuged at 3,000 rpm at low speed for 20 min, and then rinsed three times with sterilized deionized water to fully remove the medium, and then an equal amount of mycelia was added to 100 mL of sterile deionised water mixed with potato glucose liquid medium in a 1:5 ratio as a control group, and 100 ml of fig leaf extract mother liquor mixed with potato glucose liquid medium in a 1:5 ratio as a treatment group. after which the mixture was incubated at a constant temperature of 200 rpm at 28 °C in a constant temperature shaker. At 0, 0.5, 1.5, 2.5, 3.5, 4.5, 5.5, 6.5, 7.5 and 8.5 h, 1 mL of supernatant was collected and centrifuged at 3,000 rpm for 5 min by a high-speed low-temperature centrifuge to measure the conductivity of the supernatant. Afterward, the sample was replenished with sterile deionized water, and each test was repeated three times.

Determination of enzyme activity

Samples of F. equiseti that were cultured in a constant temperature shaker at 28 °C for 48 h were taken, fig leaf extract was added to potato dextrose liquid medium on an ultraclean bench, fig leaf extract was filtered through a sterile 0.20 mm microporous membrane, and an equal amount of sterile deionized water was added to the negative control group. The mycelia of the F. equiseti pathogen were removed at 0, 3, 6, 12 and 24 h and washed three times with PBS. The samples were subsequently filtered with gauze, and the water was removed. The samples were frozen in liquid nitrogen and stored at −80 °C.

To determine the MDA content, 0.1 g of mycelia was weighed, 1 mL of extraction solution was added, the homogenate was fully ground with a mortar and pestle at low temperature, and the mixture was centrifuged at 10,000 rpm and 4 °C at low temperature for 10 min. Afterward, 225 µL of the supernatant was collected to measure the MDA content in the mycelia according to the instructions of the kit by an ultraviolet spectrophotometer (Malondialdehyde test kit M0106B).

The standard curve was y = 0.0391 × −0.0076, R2 = 0.9997, x is the concentration of the standard (nmol/mL), and y is the ΔA. MDA content (nmol/g) = (ΔA + 0.0076) ÷ 0.0391 × Vti ÷ W.

Determination of protein content: First, 0.1 g of mycelia was weighed, added to 1 mL of distilled water, fully ground and homogenized with a mortar and pestle at low temperature. Afterward, the mixture was centrifuged at 12,000 rpm at 4 °C at a low temperature for 10 min. Afterward, 500 µL of the supernatant was taken, and a blank control tube was used to adjust the temperature to zero. The content of the proteins in the mycelia was measured in accordance with the instructions of the kit (Kaumas blue protein content test kit M1805B).

For the standard curve, y = 14.253 x −0.0007, R2 = 0.9997, x is the concentration of the standard (mg/mL), and y is the ΔA. Cpr (mg/g fresh weight) = 0.07 × (ΔA + 0.0007) ÷ W.

Determination of hydrogen peroxide content: Mycelia (0.15 g) were weighed and combined with 1.5 mL of extraction solution, fully ground with a mortar and pestle at low temperature until a white powder was formed. The mixture was transferred to an EP tube, 1.5 mL of the extract was centrifuged in by a high-speed low-temperature centrifuge at 10,000 rpm at 4 °C for 10 min, and then 1,000 µL of the supernatant was collected for H2O2 content determination in the mycelium according to the instruction manual of the kit (Hydrogen peroxide test kit M0107B).

The standard curve was y=0.6674 × −0.0155, R2 = 0.9997, × is the concentration of the standard (µµmol/mL), and y is the ΔA. H2O2content (µµmol/mL) = (ΔA + 0.0155) ÷ 0.6674 × Vti ÷ W.

Determination of protective enzyme activity

Superoxide dismutase (SOD) activity: Mycelia (0.1 g) were added to 1 mL of extraction solution, a mortar and pestle was used to fully grind the mixture to white powder, and the homogenate was centrifuged in a high-speed low-temperature centrifuge at 12,000 rpm and 4 °C for 10 min. A total of 50 mL of supernatant was collected and processed according to the instructions of the kit(Superoxide dismutase test kit M0102B) for micro adjustment to measure SOD activity.

SOD activity (U/g fresh weight) = 20 × inhibition rate ÷ (1 − inhibition rate) ÷ W.

Peroxidase (POD) activity: Mycelia (0.1 g) were weighed, 1 mL of extraction solution was added, the mixture was ground to a white powder with a mortar and pestle at low temperature, and the mixture was centrifuged in a high-speed low-temperature centrifuge at 12000 rpm and 4 °C for 10 min. Afterward, 50 mL of the supernatant was collected to measure the POD activity according to the manufacturer instructions (Peroxidase test kit M0105B).

POD content (U/g fresh weight) = ΔA × V1 ÷ (W × V2 ÷ V3) ÷ 1 ÷ T.

Catalase (CAT) activity: Mycelia (0.1 g) was combined with 1 mL of extraction solution, fully ground with a mortar and pestle at low temperature until a white powder formed, and centrifuged in a high-speed low-temperature centrifuge at 12,000 rpm and 4 °C for 10 min. Supernatant (35 mL) was subjected to CAT activity determination according to the manufacturer instructions (Catalase test kit M0103B).

CAT content (µµmol/min/g fresh weight) = [ΔA × Total V counter ÷ (ℇ × d) × 106] ÷ (W × V sample ÷ Total V sample) ÷ T = 678 × ΔA ÷ W.

Effect of fig leaf extract on the lesion diameter of damaged inoculated fruit

The spore suspension was prepared according to the methods of Bi, Tian & Guo (2006). The PDA medium containing F. equiseti was cultured at 28 °C for 7 d. Approximately 10 mL of sterile water containing 0.05% Tween 20 was added by volume, and the spores of F. equiseti on the PDA medium were scraped off with a glass rod transferred into 50 mL triangular flasks, oscillated on a vortex mixer for 15 s, and subsequently filtered through double-layered gauze. The filtrate was counted on a hematology counting plate to calculate the concentration of the spore suspension, which was subsequently diluted to 1 × 106 spores/mL with sterile water.

Fruit damage was induced via inoculation according to the methods of Bi, Tian & Guo (2006). The fruits were first rinsed with tap water, soaked in 2% sodium hypochlorite solution by volume for 1 min, rinsed with tap water and dried at room temperature (22 ± 2 °C). After surface disinfection of the fruit inoculation site with 70% ethanol solution by volume, five holes (3 mm deep) were punched uniformly in the equatorial part of the fruit with a sterilized hole punch (6 mm in diameter). The holes were inoculated with 20 µL of the above spore suspension, inoculated with sterile water as the control group, dry and store in cling film sealed at room temperature. The diameter of the spots was measured at 3, 6 and 9 days after inoculation. Three fruits were used for each treatment at each time point, and the results were replicated three times.

LC—MS detection of secondary metabolites in fig leaf extracts

Sample preparation and extraction: Take out the sample from the −80 °C refrigerator and thaw it on ice, vortex for 10 s to mix well. Take 100 μL fig leaf extract of the sample, add it to the 2.0 mL centrifuge tube. Add 100 μL of 70% methanol internal standard extract, vortex for 15 min. Centrifuge at 12,000 r/min for 3 min at 4 °C.Pipette the supernatant, filter it with a microporous membrane (0.22 μm), and store it in an injection vials for LC-MS/MS detection, and the samples were subsequently sent to Wuhan Mavi Metabolic Technology Service Co.

UPLC Conditions: Agilent SB-C18 (1.8 µm, 2.1 mm * 100 mm): The mobile phase was consisted of solvent A, pure water with 0.1% formic acid, and solvent B, acetonitrile with 0.1% formic acid. Sample measurements were performed with a gradient program that employed the starting conditions of 95% A, 5% B. Within 9 min, a linear gradient to 5% A, 95% B was programmed, and a composition of 5% A, 95% B was kept for 1 min. Subsequently, a composition of 95% A, 5.0% B was adjusted within 1.1 min and kept for 2.9 min. The flow velocity was set as 0.35 mL per minute; The column oven was set to 40 °C; The injection volume was 2 μL. The effluent was alternatively connected to an ESI-triple quadrupole-linear ion trap (QTRAP)-MS.

ESI-Q TRAP-MS/MS: The ESI source operation parameters were as follows: source temperature 500 °C; ion spray voltage (IS) 5,500 V (positive ion mode)/−4,500 V (negative ion mode); ion source gas I (GSI), gas II (GSII), curtain gas (CUR) were set at 50, 60, and 25 psi, respectively; the collision-activated dissociation (CAD) was high. QQQ scans were acquired as MRM experiments with collision gas (nitrogen) set to medium. DP (declustering potential) and CE (collision energy) for individual MRM transitions was done with further DP and CE optimization. A specific set of MRM transitions were monitored for each period according to the metabolites eluted within this period.

Data processing

The measured values of each indicator were statistically analyzed and plotted using Origin 8.5. The data were subjected to analysis of variance (ANOVA) using SPSS 17.0 data processing software, and the significance of the differences was analyzed using Duncan’s multiple comparisons (P < 0.05 indicates significant differences).

Results

Effect of fig leaf extracts on colony diameter

The fig leaf extract showed different inhibitory effects on all postharvest fungal diseases of melon plants. As shown in Fig. 1, fig leaf extract had more obvious inhibitory effects on H-3, H-5, H-6, and No. 1 than on other strains, for which the inhibition rates were 46.80%, 42.88%, 45.31%, and 42.34%, respectively. The difference reached a highly significant level (P < 0.01), whereas the inhibitory effects on H-10 and two fungi were inhibited, but the difference was not significant.

Figure 1 Determination of the bacteriostatic activity of fig leaf extract against the postharvest pathogen Fusarium in melons.

Effect of fig leaf extract on the morphology of F. equiseti cells

Scanning electron microscopy revealed that the surface of the mycelia of F. equiseti in the control group had fine folds, the mycelia were cylindrical and had a smooth surface and uniform thickness, and the tip of the mycelia was a full semicircle (Figs. 2A and 2B); however, after 24 h of treatment with fig leaf extract, the surface of the mycelia appeared to have more folds, exhibiting a notable puckering phenomenon accompanied by a small amount of leakage of cytoplasmic substances (Figs. 2C and 2D). Transmission electron microscopy revealed that in the control group, the cell wall and cell membrane of the fungal hyphae were clear and structurally intact, the interstitium was uniformly thin and clear, the intracellular cytoplasm was structurally intact, and the nuclei were rounded in the center of the cell, with vacuoles of varying sizes in the vicinity (Figs. 2E–2G); however, in the control group, after treatment with fig leaf extract for 24 h (Figs. 2H–2J), the cell membrane was not smooth, and the interstitium was not smooth, exhibiting a small amount of cytoplasmic leakage (Figs. 2C and 2D). The cell membrane edges were not smooth, the intercellular plasm was enlarged, the surface of the cell wall was notably rough, leakage of intracellular cytoplasm occurred, the nucleoplasm was condensed, the nucleus was fragmented, and apoptotic vesicles were formed, among other morphological changes.

Figure 2 Scanning electron microscopic images of different treatments on F. equiseti in melon.

(A and B) Control; (C and D) fig leaf extract treatment of F. equiseti on melon after 24 h; (E–G) control; (H–J) fig leaf extract treatment of F. equiseti after 24 h.

Therefore, the results of scanning electron microscopy tests revealed that fig leaf extract can increase the extravasation of substances within the cell membrane, resulting in the collapse of the mycelial cell wall and cell membrane due to the leakage of large amounts of cytoplasmic debris, after which the mycelia undergo puckering and nonuniform swelling.

PI and FAD staining to determine the effect of fig leaf extracts on F. equiseti cell membrane integrity

PI is a nuclear staining reagent that stains DNA and can be used to verify that cell membranes are not damaged; if red fluorescence is observed under a fluorescence microscope after staining, the plasma membrane is not intact. Similarly, FAD staining can be used to verify whether the cell membrane is intact; if green fluorescence is observed under a fluorescence microscope, the cytoplasmic membrane is intact. As shown in Fig. 3, the number of spores in the control treatment was significantly greater than that in the fig leaf extract treatment (Figs. 3A and 3D), the number of spores with red fluorescence in the control treatment was significantly lower than that in the fig leaf extract treatment after PI staining (Figs. 3B and 3E), and the number of spores with green fluorescence in the control treatment was significantly greater than that in the fig leaf extract treatment after FAD staining (Figs. 3C and 3F); these findings indicate that fig leaf extract treatment damaged the integrity of the F. equiseti cell membrane.

Figure 3 Effect of fig leaf extract on F. equiseti cell membrane integrity.

(A) Microscopic photograph of the control group; (B) fluorescence microscopic photograph of the control group after PI staining; (C) fluorescence microscopic photograph of the control group after FAD staining; (D) microscopic photograph of the fig leaf extract-treated group; (E) fluorescence microscopic photograph of the PI staining of the treated group; and (F) fluorescence microscopic photograph of the FAD staining of the treated group.

Effect of fig leaf extract on F. equiseti mycelial cell membrane permeability and MDA content

As shown in Fig. 4, the changes in cell membrane permeability and MDA content of F. equiseti mycelia in both the control and treatment groups showed a gradual increase. The cell membrane permeability increased more quickly in the treated group than in the control group, and there was a transient and rapid increase in the cell membrane permeability in the control group from 23.33 ± 2.63 to 36.50 ± 1.46 µµs/cm in 0–1.5 h. This difference may be attributed to the fact that after the addition of deionized water to the mycelia of the pathogenic fungus F. equiseti, some of the intracellular ions of the fungus were exuded into the solution from the cell due to the change in osmotic pressure. Some of the fungal intracellular ions leaked from the cell into the solution due to the change in osmotic pressure, and the extracellular conductivity generally maintained an equilibrium state from 2.5 to 8.5 h, which was 39.57 ± 0.52 µµs/cm at 8.5 h. In contrast, the cell membrane permeability of the treated group increased rapidly from 24.33 ± 6.33 µµs/cm to 47.33 ± 0.89 µs/cm in the period from 0 to 2.5 h and then increased at a more gentle rate in the period of 3.5–8.5 h, from 48.07 ± 1.5 µs/cm to 47.33 ± 0.89 µs/cm and from 48.07 ± 1.51 µs/cm to 50.07 ± 1.04 µs/cm, and the differences were all significant (P < 0.05) after 0.5 h. The MDA content also increased faster in the treated group than in the control group, with the MDA content in the control group increasing slowly from 2.84 ± 0.28 mmol/g to 4.08 ± 0.29 mmol/g; however, in the treated group, the MDA content increased rapidly from 2.89 ± 0.32 mmol/g to 6.22 ± 0.65 mmol/g. The difference reached a highly significant level (P < 0.01). The increase in MDA content after 24 h was 43.41% and 115.04% in the control and treated groups, respectively. Taken together, these findings indicate that the cell membrane of F. equiseti mycelia was disrupted by treatment with fig leaf extract, which resulted in spillage of intracellular fluids, increased conductivity in the extracellular fluid, and a significant increase in MDA content, which indicated lipid peroxidation in the cell membrane of the fungal tissues.

Figure 4 Effect of fig leaf extract on the in vitro conductivity of mycelia of F. equiseti.

Effect of fig leaf extract on the protein content of F. equiseti mycelia

As shown in Fig. 5, the protein content in the mycelia of the control and treatment groups exhibited opposite trends, with the protein content of the control group showing a gradual increase and the protein content of the treatment group showing a gradual decrease. The protein content in the mycelia of the control group increased from 0.283 ± 0.009 to 0.316 ± 0.008 mg/mL, while the protein content in the mycelia of the treated group decreased from 0.281 ± 0.006 to 0.264 ± 0.004 mg/mL, exhibiting a significant difference (P < 0.05) after 6 h of treatment. These findings indicate that fig leaf extract has a significant inhibitory effect on the soluble protein content in F. equiseti mycelia.

Figure 5 Effect of fig leaf extract on the protein content of F. equiseti mycelia.

Effect of fig leaf extract on reactive oxygen species (ROS) levels and antioxidant enzyme activities in F. equiseti mycelia

H2O2 is an important reactive oxygen species, and its content can reflect changes in ROS in mycelia. As shown in Fig. 6A, the changes in H2O2 concentration in the mycelia of the control and treatment groups were similar and showed a gradual increase. The H2O2 concentration in the control group slowly increased from 4.24 ± 0.30 mmol/mg prot to 5.25 ± 0.32 mmol/mg prot, whereas that in the treatment group increased from 4.75 ± 0.17 to 7.84 ± 0.08 mmol/mg prot, and the H2O2 concentration in the control group and the treatment group increased from 4.75 ± 0.17 to 7.84 ± 0.08 mmol/mg prot by 24 h. The rates of increase in H2O2 content were 23.99% and 65.27% in the control and treated groups, respectively. The rate of increase in H2O2 content in the treated group was greater than that in the control group, and the differences reached highly significant levels (P < 0.01) after 3 h of treatment. These findings suggested that fig leaf extract can induce excessive production of H2O2 in the mycelia of F. equiseti, thus disrupting the normal growth of Fusarium.

Figure 6 Effect of fig leaf extract on H2O2 production (A) and SOD (B), POD (C), and CAT (D) activities in F. equiseti.

CAT, POD and SOD are important antioxidant enzymes in fungi that can scavenge oxygen radicals in the body. The activities of these three enzymes exhibited similar trends, all of which tended to increase first and then decrease. The SOD activity in the treated group increased rapidly from 2.80 ± 0.29 to 3.88 ± 0.30 U/g prot within 0–3 h and then decreased slowly to 2.48 ± 0.09 U/g prot within 24 h. The difference in SOD activity between the control group and the treated group was significant (P<0.05) at 3 h, and the SOD activity increased by 38.60% (Fig. 6B). Similarly, the POD activity in the treated group increased rapidly from 0.667 ± 0.058 to 2.8 ± 0.2 U/g prot from 0–3 h. At 3 h, the difference between the control and treated groups was significant (P < 0.05), and the POD activity increased by 320% (Fig. 6C). CAT activity in the treated group increased rapidly from 0.798 ± 0.058 to 1.328 ± 0.12 U/g prot from 0–3 h, followed by a slow decrease to 0.945 ± 0.165 U/g prot at 24 h. At 3 h, the difference between the treated and control groups was significant (P < 0.05), with an increase in CAT activity of 66.41% (Fig. 6D).

Effect of fig leaf extract on the lesion diameter of damaged inoculated fruit

As shown in Fig. 7, the effects of F. equiseti inoculation and suppression by fig leaf extract were obvious for two melon varieties, Xizhou Mi 17 and Huangmeng Cui; the differences between the control group and the treatment group was significant (P < 0.05) at 6 and 3 days of inoculation, respectively; and the suppression rates reached 29.77% and 8.82%, respectively. With the prolongation of treatment, the effect of fig leaf extract did not weaken, the differences between the control group and the treatment group reached significant levels (P < 0.05) at 9 and 6 days after inoculation for Xizhou Mi 17 and Huangmeng Cui, respectively, and the inhibition rates reached 21.62% and 10.91%, respectively. These findings indicate that fig leaf extract is effective at inhibiting F. equiseti in postharvest melon and has a long duration of action.

Figure 7 Inhibitory effect of damage inoculation of different melon varieties with Fusarium spp.

(A) The performance of the Xizhou Mi 17 control and treatment inoculation at day 6; (B) the Xizhou Mi 17 inoculation at day 9; (C) Huangmeng Crisp inoculation at day 3; (D) the Huangmeng Crisp inoculation at day 6. An asterisk (*) corresponds to significance at the 5% level; two asterisks (**) correspond to significance at the 1% level.

Compositional analysis of fig leaf extracts

Using UPLC‒MS wide-target metabolomics technology, a total of 1540 metabolites were detected and analyzed in fig leaf extracts, of which amino acids and derivatives accounted for 16.04%; flavonoids, 15.45%; phenolic acids, 15%; others, 12.14%; alkaloids, 10.71%; organic acids, 7.73%; lipids, 7.47%; lignin and coumarin, 5.91%; nucleotides and derivatives, 4.61%; terpenoids, 3.18%; quinine, 1.43%; and tannins, 0.32% (Fig. 8). Flavonoids, phenolics, and alkaloids, which exhibit fungicidal activity, were present at relatively high percentages.

Figure 8 Ring diagram of secondary metabolite category composition of fig leaf extracts.

Based on the results of broad-targeted metabolomic substance analysis, the substances with the highest relative content in each classification were counted as shown in the Table 2. Among the 12 substance classifications, the compound with the highest relative content of flavonoids is 1,3,7,8-tetrahydroxy-2-prenylxanthone, whose chemical formula is C18H16O6; the one with the highest relative expression in alkaloids is 8-hydroxyquinoline, whose chemical formula is C9H7NO; the one with the highest relative expression in organic acids is azelaic acid, its molecular formula is C9H16O4; the compound with the highest relative expression in lignin and coumarin is 7-Hydroxycoumarin, its chemical formula is C9H6O3; the compound with the highest relative expression in quinone compounds is rheic acid, its chemical formula is C15H8O6, and all of the above compounds have been reported it has antibacterial, anti-inflammatory, anticancer, antiviral and antioxidant effects.

Table 2 Substances with the highest relative content in the classification of 12 substances.

Class	Compounds	Formula	Ionization model	Relative content		
Alkaloids	8-hydroxyquinoline	C9H7NO	[M+H]+	2.13E+07 ± 1.69E+06		
Amino acids and derivatives	L-Valine	C5H11NO2	[M+H]+	8.29E+06 ± 3.86E+05		
Flavonoids	1,3,7,8-tetrahydroxy-2-prenylxanthone	C18H16O6	[M+H]+	3.72E+03 ± 1.16E+03		
Lignans and coumarins	7-Hydroxycoumarin;Umbelliferone	C9H6O3	[M−H]−	6.08E+05 ± 3.33E+04		
Lipids	9-Hydroxy-12-oxo-15(Z)-octadecenoic acid*	C18H32O4	[M−H]−	1.93E+06 ± 3.62E+04		
Nucleotides and derivatives	N-(1-Deoxy-1-fructosyl)Phenylalanine	C15H21NO7	[M−H]−	4.58E+06 ± 5.18E+05		
Organic acids	Azelaic acid	C9H16O4	[M−H]−	8.04E+05 ± 9.31E+03		
Others	5-Hydroxy-8,8-dimethyl-1,3,4,4a,4b,5,6,7,8a,9-decahydrophenanthren-2-one	C16H24O2	[M+H]+	6.05E+06 ± 1.11E+05		
Phenolic acids	β-(3-Hydroxy-4,5-dimethoxyphenyl)-O-(6′-O-galloyl)-glucopyranoside	C21H24O13	[M−H]−	5.21E+06 ± 2.58E+05		
Quinones	Rheic acid	C15H8O6	[M−H]−	2.62E+05 ± 5.87E+03		
Tannins	3,3′-Di-O-Methylellagic acid 4′-glucoside	C22H20O13	[M−H]−	1.79E+05 ± 3.15E+04		
Terpenoids	Cis-abienol	C20H34O	[M−H]−	7.78E+04 ± 8.53E+02		

Discussion

The cell wall and cell membrane play crucial roles in maintaining the morphology of fungi, and plant-derived flavonoids can destroy the integrity and permeability of the cell wall membrane, thus disrupting the cellular morphology of microorganisms (Cushnie & Lamb, 2011). In the present study, significant changes in the morphology and ultrastructure of F. equiseti after treatment with fig leaf extract were observed via electron microscopy, which showed leakage of cell contents, crumpling and even collapse of hyphae, and disruption of the integrity of the cell membrane, which led to an increase in the conductivity of the extracellular fluid. PI-FAD double staining also confirmed the disruption of F. equiseti cell membrane integrity and a decrease in spore activity. Wang et al. (2017) reported that Fusarium rotundum mycelia were severely curved, wrinkled and collapsed after treatment with carvacrol and eugenol, and PI staining revealed that the treatment disrupted the integrity and permeability of the F. rotundum cell membranes, which was in agreement with the results of the present study. Thus, fig leaf extract inhibited fungal growth by disrupting Fusarium cell membrane permeability.

Proteins, as the basic material basis of life activities, play an important physiological role in cells, and a reduction in protein content affects the normal physiological function of cells (Yin et al., 2020). Moreover, protein leakage is one of the signs that the integrity of the cell membrane is disrupted (Bajpai, Sharma & Baek, 2013). In this study, the protein content in the mycelia of the control and treatment groups showed opposite trends, with a gradual decrease in the protein content in the mycelia of the treatment group, which demonstrated that the synthesis of fungal proteins was inhibited. Liu et al. (2018) demonstrated that an aqueous solution of total ginseng stem and leaf saponins at 10 mg/mL inhibited the synthesis of Fusarium rotundifolium proteins, which led to an inhibition of mycelial growth. Therefore, fig leaf extract can achieve fungi inhibition by inhibiting mycelial protein synthesis.

Membrane peroxidation and oxygen radical reactions maintain a dynamic equilibrium state in fungi to maintain the normal metabolic processes of fungal cells, and once this equilibrium is disturbed, the cell membrane of fungi can be damaged, resulting in cellular injury (Yan, 2015). Many studies have shown that membrane peroxidation is one of the main reasons for the increase in cell membrane permeability (Chowhan, Singh & Batish, 2013; Duma et al., 2012). MDA is an important metabolite in the process of biomembrane peroxidation, and its content in mycelia can reflect the degree of damage to tissue membranes (Guo et al., 2017). In the present study, the content of MDA in Fusarium mycelium increased significantly after treatment with fig leaf extract, and at the same time, it induced the production of excessive H2O2 in the mycelia, and the contents of both malondialdehyde and H2O2 showed a tendency to increase gradually. It has been shown that when tissues produce excessive amounts of reactive oxygen species, they cause oxidative damage to proteins, DNA, membrane plasmids and other cellular components (Martinez-Finley, Gavin & Gunter, 2013). Therefore a sudden increase in reactive oxygen species content in mycelium is an important indication of the occurrence of lipid peroxidation process (Ballester, Lafuente & Gonzalez-Candelas, 2006), which is also consistent with the results of the present study.

SOD, CAT and POD are important antioxidant enzymes in fungi that can scavenge oxygen radicals in the body, and changes in the activities of these three enzymes can be used to reflect the process of membranous peroxidation in fungal mycelia (Qu, Li & Cheng, 2017). In this study, the activities of these three protective enzymes tended to increase and then decrease, indicating that in the early stage of fig leaf extract treatment, the defense system of fungal tissues produces many protective enzymes to resist the oxidative damage to tissues caused by excessive ROS; however, as the fig leaf extract processing time increased, the activities of the three enzymes decreased due to the intensification of fungal tissue damage, This is consistent with the findings of Jiang et al. (2023).

Plant-derived natural products mainly refer to the secondary metabolites of plants, these metabolites are widely available in nature, are extremely diverse, and include many types of effective antifungal agents. In the present study, the secondary metabolites in fig leaf extracts were analyzed via UPLC‒MS through a broad-targeted metabolomics technique, and it was found that there was a high percentage of fungiostatically active compounds, such as flavonoids (15.45%), phenolic acids (15%), and alkaloids (10.71%). This finding is in agreement with the results of Su et al. (2023) on the major bioactive constituents of fig leaves. It has been shown that the flavonoid extracts 2,5-dicyclopentenyl cyclopentanone from marigold can inhibit the growth of watermelon wilt fungus (Du, Liu & Sun, 2017). Flavonoids from bitter ginseng have inhibitory effects on bacteria and fungi (Hadadi, Nematzaden & Ghahari, 2020). In this study, The compound with the highest relative flavonoid content is 1,3,7,8-tetrahydroxy-2-prenylxanthone, which has been reported to have antifungal, anti-inflammatory, anticancer, antiviral, and antioxidant effects (Wen et al., 2021). while the alkaloid with the highest relative expression is 8-hydroxyquinoline, which has been reported to react with metal ions, which has been reported to react with metal ions to form metal ion complexes as phytofungicides for the control of plant diseases, and is an excellent heterocyclic skeletal structure for the synthesis of drugs (Gao et al., 2017). The highest relative expression of organic acids is Azelaic acid, which is a signalling component of systemic acquired resistance (SAR) in plants (Wang et al., 2022). The highest relative content of lignans and coumarins is 7-hydroxycoumarins, whose derivatives have a wide range of biological activities such as anti-inflammatory, antioxidant, antifungal, antiviral, and antiproliferative effects, and thus have great potential for therapeutic applications. Anita et al. (2023) which supported the fungisatatic effect of fig leaf extract.

Conclusions

Fig leaf extract treatment disrupted the integrity of the Fusarium cell membrane, increased conductivity, disrupted protein synthesis, increased the leakage of cell contents and promoted the accumulation of MDA, which exacerbated the degree of cell membrane damage, induced changes in antioxidant enzyme activities in the organism, and disrupted the balance of reactive oxygen species metabolism. These changes led to the inhibition of fungal growth. The authenticity and reliability of the inhibitory effect of fig leaf extract on Fusarium was confirmed by the damage inoculation test of Fusarium spores. High expression of fungicidal actives such as 1,3,7,8-tetrahydroxy-2-prenylxanthone, 8-hydroxyquinoline, azelaic acid and 7-hydroxycoumarins was found by analysing the composition of fig leaf extracts, further confirming the possibility of fungicidal properties of fig leaf extracts. Therefore, this study revealed the inhibitory effect of fig leaf extract on Fusarium and the underlying mechanism and provides a theoretical basis for the use of fig leaf extract as a natural plant-derived fungicide.

Supplemental Information

Supplemental Information 1 Raw Data.

Supplemental Information 2 All samples GC MS data.

Additional Information and Declarations

Competing Interests

Author Contributions

Data Availability

The authors declare that they have no competing interests.

Jun Yao conceived and designed the experiments, performed the experiments, analyzed the data, prepared figures and/or tables, authored or reviewed drafts of the article, and approved the final draft.

Xinli Geng conceived and designed the experiments, authored or reviewed drafts of the article, and approved the final draft.

Heyun Zheng performed the experiments, analyzed the data, prepared figures and/or tables, and approved the final draft.

Zhiwei Wang performed the experiments, analyzed the data, prepared figures and/or tables, and approved the final draft.

Cuihuan Zhang performed the experiments, analyzed the data, prepared figures and/or tables, and approved the final draft.

Jing Li performed the experiments, prepared figures and/or tables, and approved the final draft.

Zaituna Maimaiti performed the experiments, prepared figures and/or tables, and approved the final draft.

Yong Qin conceived and designed the experiments, authored or reviewed drafts of the article, and approved the final draft.

The following information was supplied regarding data availability:

The raw measurements are available in the Supplemental Files.

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
