# Peer review of "Study on the inhibitory mechanism of fig leaf extract against postharvest Fusarium in melon"

_PeerJ, doi:10.7717/peerj.17654_

## Round 0.1 · original submission · Major Revisions

Dear Authors,

Thank you for submitting your manuscript. We have completed the review of your manuscript and a summary is appended below. The reviewers have observed some serious issues and recommend major revisions, required before publication can be considered. If you can address all reviewer comments in full, I invite you to resubmit your manuscript. We ask that you respond to each reviewer comment by either outlining how the criticism was addressed in the revised manuscript or by providing a rebuttal to the criticism.

**Language Note:** The review process has identified that the English language must be improved. PeerJ can provide language editing services - please contact us at [email protected] for pricing (be sure to provide your manuscript number and title). Alternatively, you should make your own arrangements to improve the language quality and provide details in your response letter. – PeerJ Staff

Reviewer 1 ·

Basic reporting

Why was only F. equiseti used when the number of fungi isolated was high?
Some data in the article must be entered correctly, ''1x106''.
86- Why bacteriostatic property?
Could you please explain?
120- How were bacteria removed?
148- What does "pathogenic bacteria" mean in fungus?
226- Under what conditions was the experiment set up? If it was established under room conditions, were there any other developments other than Fusarium?
Has the toxicity of the plant extract been examined as a positive control in fruit trials?
249- Why were only F. equiseti cells examined from the isolated samples?

Experimental design

It was a good study with a wide scope of research.

Validity of the findings

No comment.

Annotated reviews are not available for download in order to protect the identity of reviewers who chose to remain anonymous.

·

Basic reporting

Observation:
1. In the abstract it is mentioned that F. equiseti was the fungus selected for this study, although in the introduction they give weight to the toxic damage by F. Oxiporum to conclude with the objective of the research. Please improve the wording of that part, since F. Oxiporum was not the objective of this work.

2. In "Test Instruments", at least enter the information of the model used.

3. In methodology it was mentioned three times "...according to the kit instructions", but it was never described what type of kit was used.

4. In the SEM images, the vision scales are not shown in all.

Experimental design

no comments

Validity of the findings

After presenting the results, there was little contrast between the findings and other research.

Additional comments

Details marked in yellow (numbered line in manuscript):
49 scientific notation
56, 70 scientific notation (throughout the document)
56, 60, 61 "space"... (review this detail throughout the document)
61, 63 "et al.," (review this detail throughout the document)
114 "...sealed, AND PLACED in a refrigerator..."
141 units of measuremment
155, 156 improve the wording of this sentence, it´s not clear.
170, units error, centrifuge units are rmp.
168 at 180 IS THE SAME???, improve the wording.
187 subscript
197 formula for SOD activity?

·

Basic reporting

Very poorly written the MS

Experimental design

Not defined properly

Validity of the findings

rejected, and resubmitted after thorough revision

Additional comments

The MS entitled " Study on the inhibitory mechanism of fig leaf extract against postharvest Fusarium in a melon" was rejected, and resubmitted after thorough revision
many mistakes in the entire MS
Sentences are very poorly written through the MS
Materials method not defined properly
The results were poorly written
Not following how to write the Scientific name of the pathogen
Many places written bacteria instead of fungi

Reviewer 4 ·

Basic reporting

Areas for Improvement:

- Technical Terminology and Precision: The manuscript occasionally misuses technical terminology, as noted with "injection bottles" instead of the correct term "injection vials." This indicates a need for a thorough review and correction of specialized terminology to ensure accuracy and professionalism.
- Clarity and Accessibility of Supplementary Materials: The manuscript lacks direct references or descriptions of supplementary materials within the text. A detailed list of the 1540 secondary metabolites identified should be provided as supplementary material, with explicit references within the manuscript to enhance accessibility and utility.
- Data Availability Statement: The manuscript does not explicitly mention how or where the raw data supporting the study's findings can be accessed. A clear statement on data availability is crucial for transparency and reproducibility.

Experimental design

Areas for Improvement:

- Detailed Description of LC-MS Methods: The manuscript provides an inadequate description of the LC-MS experiments, particularly the "broad-targeted metabolomics" study. Details such as the mass spectrometry instruments used, ionization techniques, mass analyzer type, and the rationale for choosing scheduled MRM experiments on a triple quadrupole mass spectrometer over an untargeted approach on high-resolution mass spectrometry should be clearly described.
- Control Measures: The manuscript could be enhanced by explicitly stating any control measures used in the experiments.
- Ethical Considerations: The manuscript should address any ethical considerations related to the collection of biological materials, ensuring compliance with broader scientific standards for ethical research practices.

Validity of the findings

Areas for Improvement:

- Limitations of Mass Spec-Based Identification and Quantitation: The manuscript should discuss the limitations associated with mass spec-based identification and quantification of metabolites, including ion suppression, matrix effects, and the challenge of identifying unknown compounds without high-resolution accurate mass.
- Consideration of Other Inhibitory Factors: The manuscript primarily attributes the inhibitory effect on Fusarium to peroxide species within the fig leaf extract. It should also explore other bioactive compounds or synergistic effects between metabolites that might contribute to fungicidal activity.

Additional comments

- Need for Comprehensive LC-MS Data: The authors are advised to include a full list of secondary metabolites detected in the supplemental materials, detailing metabolites and emphasizing bacteriostatic active substances like flavonoids, phenolic acids, and alkaloids.
- Expansion on Future Research Directions: The manuscript would benefit from a discussion on future research directions, including investigating the synergistic effects of metabolites and the specific bioactivity of individual compounds.

Recommendation: Considering the critical points identified, including the need for more detailed methodological descriptions, discussion of limitations, and expansion of future directions, I recommend a major revision of the manuscript. The authors should address the detailed points mentioned, particularly enhancing the description of LC-MS methodology, discussing limitations and alternative inhibitory factors, and providing comprehensive metabolite data. Given the extent of these revisions, a re-review would be beneficial to ensure that all concerns have been adequately addressed.

---

## Round 0.2 · Major Revisions

Dear Authors,

We have noticed several mistakes in different sections of your manuscript. You are advised to resubmit your improved manuscript after making necessary corrections suggested by the Reviewers.

·

Basic reporting

Despite having attended to the suggestions for changes in the writing, there are still some stylistic flaws that were previously pointed out in the different sections of it. Just a small example: On line 8, they didn't put space between words “1.College of…” and “Xinjiang,830000”. This is repeated even in the “references” section, where there is a missing or extra space between words. Furthermore, in tables and figures they neglected to maintain the same style, for example, in the axis titles. It is an interesting job but it requires that the writing fit the parameters of a scientific journal.

Experimental design

no comments

Validity of the findings

no comments

·

Basic reporting

The overall modification in the manuscript entitled “Study on the inhibitory mechanism of fig leaf extract against postharvest Fusarium in melon is accepted, but needs major revision in many places of the MS

Experimental design

Before comments few suggestion to the author
Author not taken care to prepare the MS in proper way from abstract to references
Many places it is confusing whether he worked on bacteria and fungi
Many places he has written bacteriosatatic rather then fungisatatic
Entire MS he not italicized the scientific name of the fungi viz Fusarium equiseti
The protocol described in the Material and methods in not proper way

Validity of the findings

Line 21 to 44 abstract need major revision
Line 48 to 91 many correction in the introduction part
Line 92 to 240 please check entire Material and methods, each line by line modify all the marked comments
Lines 242 to 360 rephrase the sentence and correct the all comments
Lines 361 to 391 Check al sentences
Overall some of the minor corrections marked in the original MS that should be revision before submission

Reviewer 5 ·

Basic reporting

Yao and colleagues presented a manuscript with a very interesting project, which can have potential implications in the field. However as the MS stands, it requires an extensive revision and re-structure prior to be submitted for peer review.

The MS will need to be carefully revised in order to improve the language and provide clarity of ideas and concepts.

It is important that the introduction states the current state of the art in the field and pose the big question behind this MS.

Materials and methods are poorly written and described. The authors don't need to have a session of "test instruments", instead those should appear in the main methods associated to the step in which they were used. Important to make reference to all information that can allow anyone to repeat the experiment just by following the MS. When referring to any commercial kit, it is important to mention the name and /or catalog number. Providing only the company and not the kit reference is not useful.

Figures need more detailed legends. The reader should be able to read the legend and understand the results from the figure. Currently the legends are not descriptive enough. The figures also need more details. Time of incubations, treatment. For example in figure 3, there is a lot of information that should be added to the figure and the legend. Plots on figures need to be checked, for example last time point of one of the plots on figure 4 has no error bars. What are the stats, the error bars, the replicate number, what and how was measured. As this information need to be added between figures and their legends.

Results and discussion sections need to be carefully re-structured to address scientific rigor.
Conclusion section as it is, does not bring anything to the MS and is not clear in making a proper conclusion of what are the major points taken from the results obtained.

While the MS has some poorly written sections and requires thorough revision to address many major issues, I strongly encourage the authors to re-submit a new version after re-structuring and addressing the current issues. There is potential in this data and results, but the MS needs work around it to make all those points clear and to be impactful for the scientific community.

Experimental design

The overall explanation on Basic Report already covers this section.

Validity of the findings

The overall explanation on Basic Report already covers this section.

---

## Round 0.3 · accepted · Accept

Authors have improved their manuscript. Therefore, this manuscript may be accepted for the possible publication.

·

Basic reporting

In line:
1- space in "1.College".
21, 23- correct the word "fungisatatic" by fungistatic.
48, 182, 199- no space in "Fusarium ," Attention in all document.
60 and others- space before year, ex: "(Nogueira et al.,2023)".
66 and others- comma before year "(Ferreira et al.2018)".

You should double-check the entire document to make sure the style is appropriate for the journal.

Experimental design

no comment

Validity of the findings

no comment